# Rapid Evolutionary Adaptation to Diet Composition in the Black Soldier Fly (*Hermetia illucens*)

**DOI:** 10.3390/insects14100821

**Published:** 2023-10-18

**Authors:** Anton Gligorescu, Long Chen, Kim Jensen, Neda Nasiri Moghadam, Torsten Nygaard Kristensen, Jesper Givskov Sørensen

**Affiliations:** 1Department of Biology, Section for Genetics, Ecology and Evolution, Aarhus University, Ny Munkegade 116, Building 1540, 8000 Aarhus C, Denmark; jesper.soerensen@bio.au.dk; 2Department of Animal and Veterinary Sciences, Aarhus University, Blichers Allé 20, 8830 Tjele, Denmark; long.chen@anivet.au.dk (L.C.); kj@anivet.au.dk (K.J.); 3Life Science Division, Danish Technological Institute, Kongsvangs Allé 29, 8000 Aarhus C, Denmark; nnm@teknologisk.dk; 4Department of Chemistry and Bioscience, Section for Bioscience and Engineering, Aalborg University, Fredrik Bajers Vej 7H, 9220 Aalborg E, Denmark; tnk@bio.aau.dk

**Keywords:** evolutionary adaptation, insect production, low-quality diets, single-sourced waste streams, trade-offs

## Abstract

**Simple Summary:**

The production performances of Black Soldier Fly Larvae (BSFL) are highly affected by substrate quality. Currently, multiple sources of side-streams are used to develop diets of increased complexity and quality. This increases production costs (and potentially reduces profit) and results in additional environmental impacts associated with purchasing and transporting side-streams from multiple sources. In this study, we evaluate the adaptation of BSFL to a single sourced low-quality diet and the potential costs associated with diet adaptation. We tested evolutionary responses of BSFL to a single sourced, low-quality wheat bran diet (WB) or to a high-quality chicken feed diet (CF) for several traits during 13 generations. We evaluated the cost associated with adaptation by switching the diets at generation 13. Although diet quality plays an important role, the study revealed that faster evolutionary adaptation was experienced by BSFL reared on the low-quality single sourced WB diet. No costs associated with adaptation were found in the study. Our study suggests that BSFL can be adapted to feed on low-quality single source waste streams.

**Abstract:**

Genetic adaptation of *Hermetia illucens* (BSF) to suboptimal single sourced waste streams can open new perspectives for insect production. Here, four BSF lines were maintained on a single sourced, low-quality wheat bran diet (WB) or on a high-quality chicken feed diet (CF) for 13 generations. We continuously evaluated presumed evolutionary responses in several performance traits to rearing on the two diets. Subsequently, we tested responses to interchanged diets, i.e., of larvae that had been reared on low-quality feed and tested on high-quality feed and vice versa to evaluate costs associated with adaptation to different diets. BSF were found to experience rapid adaptation to the diet composition. While performances on the WB diet were always inferior to the CF diet, the adaptive responses were stronger to the former diet. This stronger response was likely due to stronger selection pressure experienced by BSF fed on the low-quality single sourced diet. The interchanged diet experiment found no costs associated with diet adaptation, but revealed cross generational gain associated with the parental CF diet treatment. Our results revealed that BSF can rapidly respond adaptively to diet, although the mechanisms are yet to be determined. This has potential to be utilized in commercial insect breeding to produce lines tailored to specific diets.

## 1. Introduction

The quantity and quality of nutrients are important for the performance of insects, and nutritional requirements and optimal diets have been investigated and described for a range of species [1]. Optimal diets are typically complex diets composed of several ingredients in order to ensure access to both the macro- and micronutrients required by the specific species. Investigating the consequences of diet quality on insects’ performance is highly relevant for the emerging insect production industry, where the major motivation is bio-converting low-quality substrates into higher value products, primarily intended for food and feed [2,3]. A commonly farmed insect species is *Hermetia illucens* (Diptera, Stratiomyidae), also known as the Black Soldier Fly (BSF) [4]. BSF larvae (BSFL) have a desirable nutritional composition, making them highly suitable in pet food, and as a feed ingredient in aquaculture, poultry, and swine sectors [5]. Further, BSFL are generalist scavengers that can be raised on a wide range of feed sources, including simple and complex industrial side- and waste streams [5,6]. 

BSFL diets based on side- and waste streams, however, are often low-quality diets considered suboptimal for growth, reproduction, and other performance traits [7]. For example, the production time was long and larval weight was low when BSFL were fed on simple, low-quality substrates such as apple pulp, forced chicory roots, tomato leaves, or grain middling [6]. Several studies have revealed that feed conversion rate (FCR) and substrate reduction were lower when BSFL were fed on simple, low-quality substrates such as corn stover [8], fermented maize stover [9], or fruit and vegetable waste [5,8,9,10] when compared to BSFL fed on complex diets such as bakery waste [11] or commercial chicken feed [12]. One approach to prevent low performance on suboptimal diets is to combine multiple sources of side-streams to develop diets of increased complexity and quality. This approach, however, may increase production costs (and potentially reduce profit) and result in additional environmental impacts associated with purchasing and transporting side-streams from multiple sources. Another drawback is that variable diets can result in performance and product quality variation among production batches. 

An alternative approach to maintain high performance of BSF and other insect species used for production on low-quality diets is to exploit the genetic variation of a breeding stock to achieve specialist lines that are adapted to low-quality substrates or generalist lines that can maintain high performance across multiple diets. It has been documented, from studies on the vinegar fly *Drosophila melanogaster*, that genetic variation for the ability to utilize diets of low diet quality exist [13,14]. Similar genetic variation in performance on different diets was documented by revealing interactions between diet and genetic background on body nutrient composition in the cricket, *Teleogryllus commodus* [15] and on larval performances and body nutrient composition in BSFL [16]. Furthermore, juveniles of the moth *Plutella xylostella* were able to adapt to novel, low-starch diets when reared during multiple generations on such diets [17]. Such data suggest that BSF may adapt evolutionarily to low-quality diets and that selection could be a feasible strategy for optimizing commercial BSF production. Phenotypic selection has recently been used to efficiently improve larval weight, larval biomass, crude protein content, and lipid content in BSFL [18]. 

Here, we maintained four lines of BSF on either a single-sourced, low-quality diet (wheat bran) or on a high-quality diet (chicken feed) for 13 generations in order to evaluate whether the lines have responded evolutionarily to these diets. In this experimental evolution set up, we investigated the larval biomass, FCR, survival rate, frass mass, average pupa weight, and metamorphosis rate in every generation from generation 5 to 13. We hypothesized that, throughout the 13 generations, BSFL would improve phenotypic performance on the simple, low-quality diet, suggesting adaptation to this diet. Subsequently, we interchanged the diets during generation 13 to test the hypothesis that evolutionary adaptation to given diets comes at a cost when exposed to novel diets. 

## 2. Materials and Methods

### 2.1. Culture Origins and Maintenance

We used a laboratory culture of *H. illucens* established in 2017 at the Danish Technological Institute (DTI) from a batch of juveniles (N > 100,000) provided by InsectProtein IVS, Sønderborg, Denmark. The BSF culture was maintained in a climate-controlled room at 14:10 h light: dark cycle, a temperature of approximately 28 ± 1 °C (mean ± sd), and a relative humidity (RH) of approximately 60 ± 10% (mean ± sd). The BSF culture was fed on a chicken feed (CF Pacostar, DLG, Fredericia, Denmark) diet and was maintained in a large population (N > 5000 adults per generation) for >40 generations prior to this experiment. 

### 2.2. Experimental Procedures

We reared flies on two different diets for 13 generations (experimental evolution setup). We selected the two diets to represent a complex diet of high-quality (CF) and a single sourced, low-quality diet (wheat bran, WB), respectively. The CF and WB substrates were purchased commercially (DLG, Fredericia, Denmark) and mixed with water at a 1:2 mass ratio of dry feed to water and 20 mL sunflower oil (COOP, Albertslund, Denmark) per kg mixture. The nutritional composition of both diets was analyzed to determine dry matter (DM), ash, macronutrient content (crude protein, lipid and carbohydrate), and the essential amino acid profile (Table 1).

The DM and ash contents were determined by drying the samples (four replicates/diet) at 105 °C for 24 h and subsequently combusting them at 550 °C for another 24 h. The crude protein contents were estimated using the Kjeldahl method [19], and the crude lipid contents were determined using gravimetric methods, both at Eurofins Steins Laboratorium A/S (Vejen, Denmark). Subsequently, the carbohydrate was estimated by subtracting the crude protein, crude lipid, and ash content from 100%, following the Weende analysis (Dry organic matter = Carbohydrate + Protein + Lipid + Ash). The amino acid profiles of CF and WB were similarly determined by Eurofins, using the ISO 13903:2005/IC-UV standard.

We set up eight lines and maintained these lines on the CF or WB diets (four replicate lines per diet) for 13 consecutive generations (F1-F13). Throughout the experiment, each new generation was established from ca. 1600 flies per line, based on randomly harvested eggs of high magnitude (N > 20,000) (Figure 1).

Collected eggs were placed on a mosquito net (1 mm mesh) on top of a nursing crate (20 × 30 × 15 cm) containing 500 g of either CF or WB diets for a six-day nursing stage. Hereafter, two consecutive sieves (mesh size:1 and 1.4 mm) were used to sieve and sort out juveniles of similar size. This was used to synchronize the developmental stage of the individuals used. The average weight of the selected juveniles was estimated by counting and weighing two random samples of juveniles (N > 50) during each generation. In addition, six samples of medium sized juveniles (N > 250) were collected during a pilot test and used to estimate the juveniles’ dry matter content (juvenile DM = 30%, Appendix A). 

The average weight of juveniles was determined for each generation in order to estimate the weight of 2500 juveniles used to initiate each line. Subsequently, the juveniles were transferred into rearing crates (20 × 30 × 15 cm) containing 2 kg of freshly prepared CF or WB diet. The crates were maintained for circa 10 days in the climate room, until about 10% of the BSFL reached the prepupa stage, indicated by a darkening larva color. The rearing crates were harvested and the BSFL were separated from the frass using a 2 mm sieve. The total larval biomass and frass mass were estimated for each crate. Subsequently, the DM content of BSFL and frass were determined for each crate for all generations. The BSFL were returned to their corresponding crates containing frass and supplemented with additional diet (ad libitum) until most of the BSFL (>90%) entered the prepupa stage (approximately four days later). Exceptions were F6 and F7, where pooled samples of BSFL and frass were taken from CF and WB crates, and F5 and F8, where samples of frass were not collected.

The BSF prepupae were harvested and transferred into pupation crates (60 × 40 × 20 cm) containing pupation medium (frass from rearing dried at 70 °C) until they were completely immobile (approximately six days later). A sample of 50 pupae were randomly collected from each pupation crate to determine the average fresh weight (FW) of the pupa of each line. These estimates were used to calculate the weight of 1600 pupae intended for mating. The pupae from each line were transferred into insect netting bags and placed in mating cages (47.5 × 47.5 × 47.5 cm).

Once flies emerged, a fresh sugar solution (1:3 ratio of white sugar to water) was provided in a small glass containing cotton. Furthermore, egg traps consisting of a silicone egg collector (custom made at DTI) and an oviposition attractant (DTI bio-attractant), were placed on top of each mating cage at predetermined oviposition sites (4 egg traps per cage). These sites were constructed with 1.5 mm mosquito mesh allowing egg laying outside the cage. The egg traps were replaced with a fresh trap every two to three days for approximately 10 days (corresponding to 4 harvesting episodes). Usually, eggs from the second or third harvest (peak of reproduction) were used to initiate the next generation. Although eggs were collected from all lines during all generations, the eggs were only systematically weighed at F13 (Figure 1).

For establishing F13, approximately 5 g of eggs were sampled from each line (i.e., adults of F12) from the second harvest egg traps and divided into two equal egg masses (2 × 2.5 g/line). Subsequently, egg masses were placed on top of nursing crates containing 500 g of CF or WB to create four treatments of all possible combinations between parent (CFp; WBp) and interchanged (CFi: WBp on CF) and (WBi: CFp on WB) diet treatments. The nursing, rearing, pupation, and mating were conducted for both the parent and interchanged treatments as described for the previous generations (Figure 1). 

### 2.3. Trait Assessment

Assessment of traits was conducted for every generation from F5 to F13. The BSFL biomass, frass mass, feed conversion ratio (FCR), average pupa weight, and metamorphosis rate were assessed for all lines and all generations. The survival rate during rearing and egg production were only assessed during the interchanged dietary experiment in F13 (Figure 1). The BSFL biomass (g FW) per line was determined by weighing the total amount of larvae harvested from each rearing crate. Subsequently, DM content (%) was used to convert the BSFL biomass (FW) to an estimate of BSFL biomass (g DM) for each line. The frass mass (g FW) per line was determined by weighing the residual substrate from corresponding rearing crates after larvae harvest. As in the case of BSFL biomass, the frass mass (g DM) per line was determined based on the frass FW mass and frass DM content. The FCR was determined both on a fresh-to-fresh and dry-to-dry matter basis as presented in Equation (1).
FCR = (Feed (g))/((BSFL biomass (g) − Juveniles mass (g)))(1)

The average pupa weight (mg FW) per line was determined based on the weight of 50 pupae taken from corresponding crates. The metamorphosis rate (%) was calculated as percentage of empty puparia out of the total number of puparia in a random subsample (N > 50) that was taken at the end of the reproduction period when most flies were dead (after approximately 10 days). Similarly, flies were randomly sampled (N > 50) and sex ratio was determined by visual inspection. However, this trait was found to be very consistent over generations (1/1 ratio) and, thus, not further addressed. In addition, at F13, the larvae survival (%) was determined for each crate, based on the estimated number of BSFL at harvest and the estimated numbers of juveniles at the beginning of rearing. Both numbers of BSFL and juveniles were estimated based on the average weight of larvae and juveniles, respectively, and the corresponding biomass weight from each crate. Lastly, egg production (g/cage) was calculated as the total weight of eggs being harvested during the reproduction period of F13 (Figure 1).

### 2.4. Data Analysis

Data were analyzed in R v. 4.2.2 [20]. Normal distribution and homogeneity of variance of data were assessed using the Shapiro–Wilk and the Bartlett tests, respectively. 

Diet adaptation experiment: BSFL biomass (FW and DM), FCR (FW and DM), and pupa weight were modelled using linear models (R function: lm) and the terms of the model (generation, diet and interaction effect) were evaluated using ANOVA (R function: anova). Generation was considered as a continuous variable and diet as a categorical variable. Frass mass (FW and DM) and metamorphosis rate failed to fulfill the normality and/or homogeneity of variance requirements and were analyzed using a generalized linear model (GLM) with the same terms and evaluation as described above. The generation term was mean centered to allow biologically relevant interpretation of the intercepts.

Interchanged diet experiment: For each trait we applied two a priori determined pairwise comparisons using *t*-tests (CFp vs. CFi, and WBp vs. WBi) to test for responses to interchanged diets in BSFL biomass, frass mass, FCR, and pupa weight. For comparing WBp to WBi for BSFL biomass (FW), we used Welch’s *t*-tests due to unequal variances, while for the BSFL biomass (DM) and metamorphosis rate we used a Kruskal–Wallis non-parametric test, due to non-normal distribution of data. In the experiment with interchanged diets (F13), the two additional traits, larval survival and egg production, were modelled using linear models (R function: lm) and the term of the model (diet) was evaluated using one-way ANOVA (R function: anova). When relevant, this was followed by post hoc pairwise comparisons using a Tukey test (R function: TukeyHSD). 

## 3. Results

The larvae biomass (DM) significantly increased over generations (F_1,68_ = 18.51; *p* < 0.001), with the CF lines experiencing a lower increase (model estimate CF = 1.82 g/generation) compared to WB lines (model estimate WB = 3.04 g/generation) in larval mass over the nine-generation assessment. The CF diet led to higher BSFL biomass compared to the WB diet (F_1,68_ = 581.32; *p* < 0.001) and no interaction was seen between generation and diet for this trait (F_1,68_ = 1.17; *p* = 0.28; Figure 2A). Analyses of the BSFL biomass in FW corroborated these findings with the exception that the interaction between diet and generation was significant (F_1,68_ = 10.87; *p* = 0.001). The overall effect of generation was mainly attributed to an increase in BSFL biomass in the WB lines (model estimate WB = 11.03 g/generation), whereas the BSFL biomass in the CF lines did not differ across generations (model estimate CF = −0.06 g/generation) (Appendix A). For the interchanged diet experiment, the BSFL biomass (DM) was similar when CFp and CFi were compared (t_6_ = 0.83; *p* = 0.44), but moderately lower in WBp when pairwise compared to WBi (Chi_21_ = 4.1; *p* = 0.04; Figure 2B). No significant difference was observed when considering the BSFL biomass in FW, as no significant differences were observed when comparing CFp and CFi (t_6_ = 0.26; *p* =0.8) and WBp vs. WBi (t_3.3_ = 2.5; *p* = −0.08), respectively (Appendix A).

The frass mass (DM) was found to be different across generations (F_1,54_ = 18.1; *p* < 0.001), with the CF lines producing a similar frass mass (model estimate CF = −0.61 g/generation) while WB lines produced a substantially lower frass mass (model estimate WB = −21.83 g/generation) over generations, leading to a significant interaction effect between diet and generation (F_1,52_ = 16.15; *p* < 0.001). Furthermore, there was a moderately higher frass mass (DM) in the WB treatment when compared to the CF treatment (F_1,53_ = 4.11; *p* = 0.048; Figure 3A). When considering FW, there was no significant effect of generation (F_1,70_ = 0.24; *p* = 0.63). Despite this, there was a slight increase in frass mass over generation for CF lines (model estimate CF = 3.9 g/generation) and a slight decrease in frass mass for WB lines (model estimate WB = −8.81 g/generation), however, not with a statistically significant interaction effect (F_1,68_ = 1.62; *p* = 0.21). Furthermore, the CF diet resulted in lower frass mass (FW) when compared to the WB diet (F_1,69_ = 46.38; *p* < 0.001; Appendix A).

Lastly, similar frass mass (DM) was obtained on either CFp or CFi (t_6_ = −1.92; *p* = 0.10), while a moderately lower frass mass was seen on WBp when compared to WBi (t_6_ = −2.59; *p* = 0.04; Figure 3B). Similarly, when considering the FW, frass production was different between CFp and CFi (t_6_ = −8.47; *p* = 0.001), but similar between WBp and WBi (t_6_ = 0.13; *p* = 0.90; Appendix A).

The FCR (DM) was affected by generation (F_1,68_ = 19.98; *p* < 0.001), with CF lines experiencing little improvement in FCR per generation (model estimate CF = −0.03/generation) and WB lines experiencing marked improvement in FCR per generation (model estimate WB = −0.13/generation), explaining a significant interaction effect of diet by generation (F_1,68_ = 6.67; *p* = 0.01) on FCR. Furthermore, the CF diet resulted in a more efficient FCR than the WB diet (F_1,68_ = 323.73; *p* < 0.001; Figure 4A). As in the case of DM, there was an effect of generation on the FCR (FW) (F_1,68_ = 15.51; *p* < 0.001), due to low improvement in the CF lines (model estimate CF = 0.0002/generation), but substantial improvement in the WB lines (model estimate WB = −0.13/generation), also explaining the interaction effect (F_1,68_ = 15.62; *p* < 0.001) on the FCR (FW) trait. Furthermore, the CF diet resulted in better FCR than the WB diet (F_1,68_ = 201.53; *p* < 0.001; Appendix A). The FCR was similar both when DM (CFp vs. CFi (t_6_ = −0.03; *p* = 0.98) and WBp vs. WBi, (t_6_ = 0.8; *p* = 0.93; Figure 4B)) and FW (CFp vs. CFi (t_6_ = −0.34; *p* = 0.74) and WBp vs. WBi, (t_6_ = 2.06; *p* = 0.08; Appendix A)) measurements were pairwise compared.

The pupa weight was found to be influenced by generation (F_1,68_ = 86.32; *p* < 0.001) in a similar manner for both the CF (model estimate CF = 5.43 mg/generation) and WB lines (model estimate WB = 6.54 mg/generation). Furthermore, there was a diet effect (F_1,68_ = 272.95; *p* < 0.001), but no diet by generation interaction effect (F_1,68_ = 0.74; *p* = 0.39) on pupa weight (Figure 5A). The pupa weight was found to be similar across parent and interchanged diets when CFp and CFi (t_6_ = 1.03; *p* = 0.34) and WBp and WBi, (t_6_ = 2.01; *p* = 0.09) were pairwise compared (Figure 5B).

Overall, the metamorphosis rate was found to be similar across generations (F_1,62_ = 0.07; *p* = 0.79) and across diets (F_1,61_ = 0.35; *p* = 0.56). The CF lines experienced a slight decrease (model estimate CF = −0.22%/generation), while the WB lines experienced a slight increase (model estimate WB = 0.26%/generation) in metamorphosis rate over generations resulting in a significant interaction effect (F_1,60_ = 7.56; *p* < 0.01, Figure 6A). A similar metamorphosis rate was seen across parent and interchanged diets when CFp vs. CFi (Chi_21_ = 0.45; *p* = 0.49) and WBp vs. WBi, (Chi_21_ = 2.28; *p* = 0.13) were pairwise compared (Figure 6B).

The larval survival at harvest was found to be overall high (>80%) and constant across the four dietary treatments (F_3,12_ = 1.68; *p* = 0.22) A tendency of higher survival rate was seen when BSFL were fed on the WB diet, potentially as a consequence of structural benefits of this diet (Figure 7).

The egg production was found to be significantly higher when BSFL were reared on CF as compared to WB (F_3,12_ = 8.75; *p* = 0.002), with a similar, high egg production on the CFp and CFi, and a similar, lower egg production on the WBp and WBi dietary treatments, respectively (Figure 8).

## 4. Discussion

We performed an experiment exposing BSF to two diets and tested them for several performance traits across multiple generations to investigate if flies responded evolutionarily to rearing on different diets. After 12 generations on the different diets we also tested flies on the alternative diet to evaluate potential trade-offs. Our study has implications for understanding the evolutionary potential of establishing lines of BSFL adapted to single-sourced side streams. As expected, BSFL raised on a high-quality CF diet showed superior performance relative to those raised on a low-quality single sourced diet. However, beyond the marked direct effect of diet, our results also revealed signs of rapid evolutionary adaptation across 13 generations in lines maintained on the low-quality single sourced WB diet. This was achieved without a clear indication of costs or trade-offs as shown by the results of the interchanged diet experiment.

Overall, larvae reared on the CF diet had higher larval biomass (Figure 2 and Appendix A), decreased frass mass (Figure 3), more efficient FCR (Figure 4 and Appendix A), larger pupa weight (Figure 5), and higher egg production (Figure 8), but a similar metamorphosis rate (Figure 6) and survival rate (Figure 7). This clearly shows the importance of diet quality for BSFL performance, which is also supported by other findings [5,21,22]. Our results are in accordance with other studies, where single source low-quality diets resulted in lower BSFL performance traits when compared to complex and high-quality diets [6,23]. The BSFL performances and nutrient profile are known to be affected by diet quality such as macronutrients composition [6,24], protein to carbohydrate ratio [25,26], and amino acid profile [23,27]. In the present study, the WB diet was found to contain 20% less protein and 36–63% less essential amino acids than the CF diet (Table 1). This may explain the lower performance of BSFL when reared on the WB diet. The strong phenotypic response to diets was previously associated with adjusting energy allocation between growth and maintenance [28] and regulating midgut functions [29] to better exploit poor diets. This could lead to trade-offs among traits, with an expectation of maintaining performance in some traits at the expense of performance in other traits. Here, we generally observed lower performance on the low-quality diet and thus no indications of such trade-offs among the measured traits. Furthermore, other diet characteristics, such as micronutrient composition and physical characteristics (e.g., porosity, particle size, water absorption) [30], as well as the microbial community may affect BSFL performance [31,32], but this was not addressed in this study. 

We found significant responses across generations, interpreted as evolutionary adaptation in all measured performance traits, except for the metamorphosis rate (Figure 2A, Figure 3A, Figure 4A, Figure 5A and Appendix A). These responses were, as expected, markedly stronger on the WB treatment, which represented a challenging diet due to it being both novel and of low-quality. Even so, for some traits the CF lines also showed a response despite being used as feed for more than 40 generations before the experiment. It is important to mention that the rearing protocol of BSFL was changed for several parameters, e.g., the size of rearing trays, between the 40 generations of colony maintenance and the current 13 generations study. Thus, both diet regimes, to some extent, constitute novel environments, which can introduce new selection pressures. The observed data for both CF and particularly WB lines indicated rapid adaptation responses across generations. Rapid adaptation to new conditions relies on ample genetic variation and strong selection intensity. Such rapid adaptive responses to new captive conditions were seen during earlier establishments of BSFL rearing colonies [33,34]. Other studies have also revealed that genetic background as well as genotype by diet interaction affects several traits in BSFL [16,35], indicating responses to different diets might have a genetic component. Interestingly, the adaptive response per generation was more pronounced in the WB lines than in the CF lines for several traits. Thus, our results indicated that WB lines experienced a stronger selection pressure than CF lines, allowing BSFL to rapidly adapt and increase performance on single source low-quality waste streams. High genetic differentiation of BSF populations to local [36,37] and regional conditions across the globe [38] might also reflect adaptation to local conditions such as diet quality, temperature, and other environmental variables. This means that naturally occurring populations might show variation in their ability and efficiency of exploiting different low-quality diets, and that this could be further refined through experimental evolution experiments. 

The seemingly adaptive response detected across generations was not detected once the diets were interchanged. In contrast, our results from the interchanged diet experiment revealed a small loss in larval biomass (DM) and frass mass (DM) in WBp when compared to WBi (Figure 2B and Figure 3B) suggesting that BSFL experience a cross generational benefit when previously exposed to a high-quality CF diet. Such gain might be a consequence of micronutrients being passed from parents to offspring.

Only a few studies document impacts on life history traits of BSFL on rearing at different diets for multiple consecutive generations [33,34]. Most existing BSF studies are conducted during a short time of their life cycle, mostly during the rearing stage (e.g., 3rd to 6th instar) [12,16,29]. Though many good reasons (economical as well as time related) are associated with considering short-term dietary experiments in BSFL, this approach has limitations. In general, BSFL are nursing on high-quality feed for 5–6 days before being placed on the dietary treatment for rearing; this might enhance larval performance and partially shade the effect of low-quality diets during the experiment. Similarly, when considering 1-generational dietary experiments, the responses in performance traits might also be affected by maternal effects, as illustrated by the results of larval biomass and frass weight of the interchanged diet experiment [39].

Exploring adaptive responses in BSFL has strong implications for the production of insects for food and feed. Such adaptative responses associated with environmental or genetic components can result in enhanced performance on low-quality industrial by-products. This can contribute to the green transition of feed and food production.

Our experiment suggests a rapid adaptation response to low-quality WB diet. When using genetic improvement caution should be taken when developing specialized lines, since strong selection may reduce genetic variation within lines, lead to inbreeding [40], and may come with a cost and reduce other performance traits [41]. Future studies integrating classical breeding, genetic information, and phenotypic evaluation can provide an avenue to explore the potential for improving performance on low-quality diets in insect production. Similarly, new studies where candidates are not selected based on phenotypes but on breeding values (the genetic merit of an individual) will likely be more efficient, leading to faster selection responses; this also allows for control of inbreeding rates and simultaneous selection on multiple traits. Traits related to performance on low-quality diets are likely highly multi-factorial and covered by many genes with small effect. Thus, selection responses are expected to be slow and other studies on important model and agricultural species have revealed continued responses to selection and no signs of reaching plateaus even after >100 generations of selection [42,43].

## 5. Conclusions

Overall, the CF diet positively affected most traits when compared to the WB diet indicating a strong impact of diet quality. However, feeding the BSFL on single source, low-quality WB diet during 13 consecutive generations was found to generate a seemingly rapid evolutionary response, suggesting that BSFL did adapt to the WB diet. To our surprise, shifting from CF to WB diet revealed cross generational effects associated with nutrient transfer from parent to offspring but with a rapid loss of the assumed genetic response. The mechanism behind the improvement seen during the experiment is yet to be revealed and understood. Regardless, our results suggest that both environment and genetic changes play an important role in BSF responses to diet quality.

## Figures and Tables

**Figure 1 insects-14-00821-f001:**
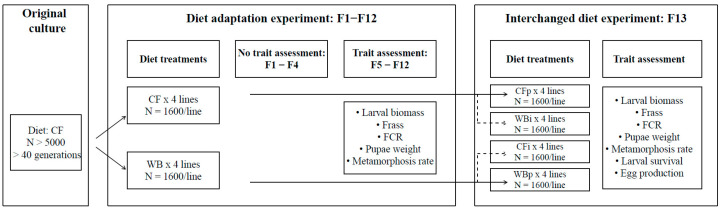
Flowchart of experimental procedure. The flow chart illustrates the dietary treatments, number for lines (replicates) and number of adult flies for: the original culture; the diet adaptation experiment composed of a pre-assessment period (F1–F4) and a trait assessment period (F5–F13); and the interchanged dietary experiment carried during F13. For this latter part, suffix “i” refers to interchanged (novel) diets and suffix “p” to parental (original) diets. Thus “WBi” refers to animals raised on the WB diet, after being raised for 12 generations on the CF diet.

**Figure 2 insects-14-00821-f002:**
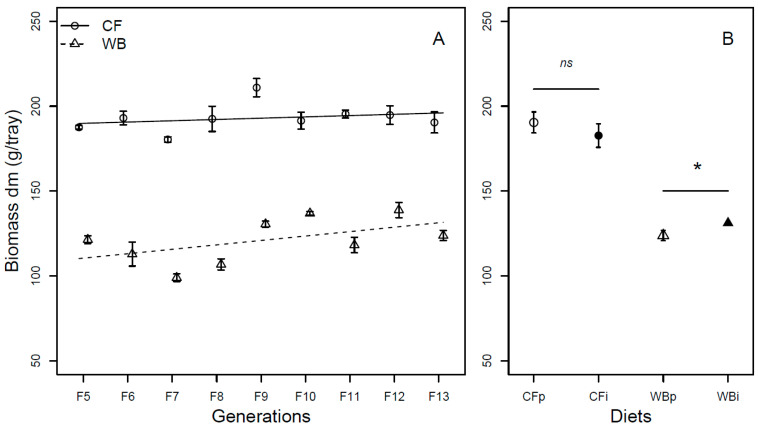
Biomass (mean ± sem) of BSFL (g/trays, DM) reared on high-quality CF diet and single-sourced, low-quality WB diet. Panel A: Responses to multiple subsequent generations. The solid black (CF) and dashed (WB) lines indicate the fitted model of the data (**A**). Panel B: Responses to the interchanged dietary experiment in F13 (**B**), where the parent dietary treatments CFp and WBp were compared to the interchanged dietary treatments CFi and WBi. Thus, “WBi” refers to animals raised on the WB diet after being raised for 12 generations on the CF diet. Lines, indicate pairwise comparisons showing no significant (ns) and significant (*) differences (*p* < 0.05).

**Figure 3 insects-14-00821-f003:**
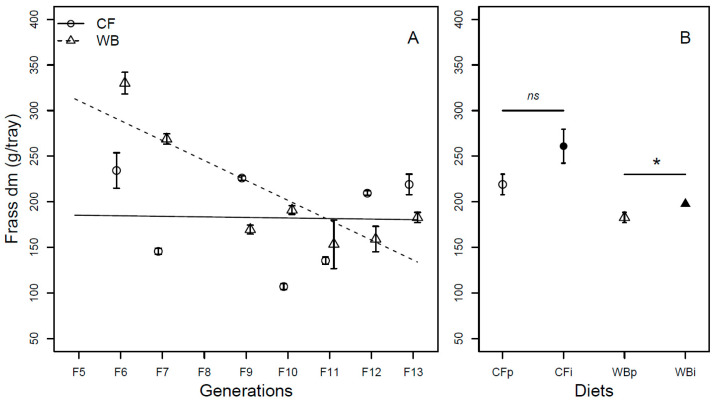
Frass weight (mean ± sem) from BSFL (g/trays, DM) reared on high-quality CF diet and single-sourced, low-quality WB diet. Panel A: Responses to multiple subsequent generations. The solid black (CF) and dashed (WB) lines indicate the fitted model of the data (**A**). Panel B: Responses to the interchanged dietary experiment in F13 (**B**), where the parent dietary treatments CFp and WBp were compared to the interchanged dietary treatments CFi and WBi. Thus, “WBi” refers to animals raised on the WB diet after being raised for 12 generations on the CF diet. Lines, indicate pairwise comparisons showing no significant (ns) and significant (*) differences (*p* < 0.05).

**Figure 4 insects-14-00821-f004:**
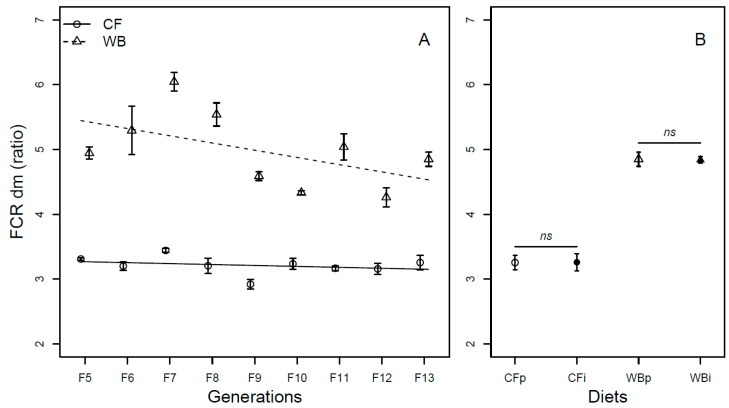
Feed conversion rate (mean ± sem) of BSFL (ratio, DM) reared on high-quality CF diet and single-sourced, low-quality WB diet. Panel A: Responses to multiple subsequent generations. The solid black (CF) and dashed (WB) lines indicate the fitted model of the data (**A**). Panel B: Responses to the interchanged dietary experiment in F13 (**B**), where the parent dietary treatments CFp and WBp were compared to the interchanged dietary treatments CFi and WBi. Thus, “WBi” refers to animals raised on the WB diet after being raised for 12 generations on the CF diet. Lines, indicate pairwise comparisons showing no significant (ns) differences.

**Figure 5 insects-14-00821-f005:**
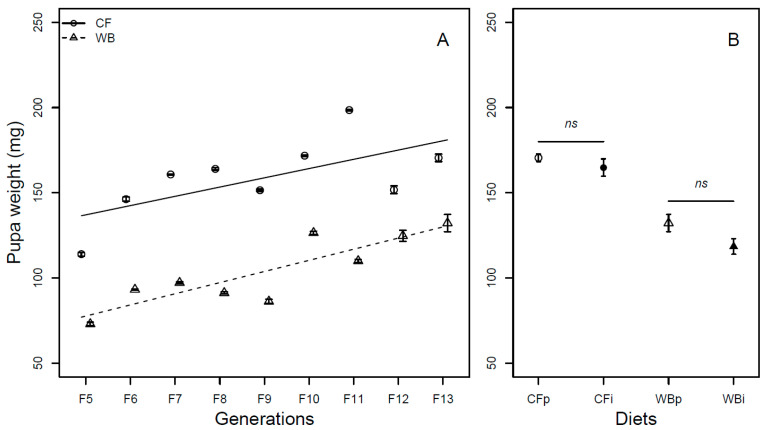
Pupa weight (mean ± sem) of BSFL (mg/pupa, FW) reared on high-quality CF diet and single-sourced, low-quality WB diet. Panel A: Responses to multiple subsequent generations. The solid black (CF) and dashed (WB) lines indicate the fitted model of the data (**A**). Panel B: Responses to the interchanged dietary experiment in F13 (**B**), where the parent dietary treatments CFp and WBp were compared to the interchanged dietary treatments CFi and WBi. Thus, “WBi” refers to animals raised on the WB diet after being raised for 12 generations on the CF diet. Lines, indicate pairwise comparisons showing no significant (ns) differences.

**Figure 6 insects-14-00821-f006:**
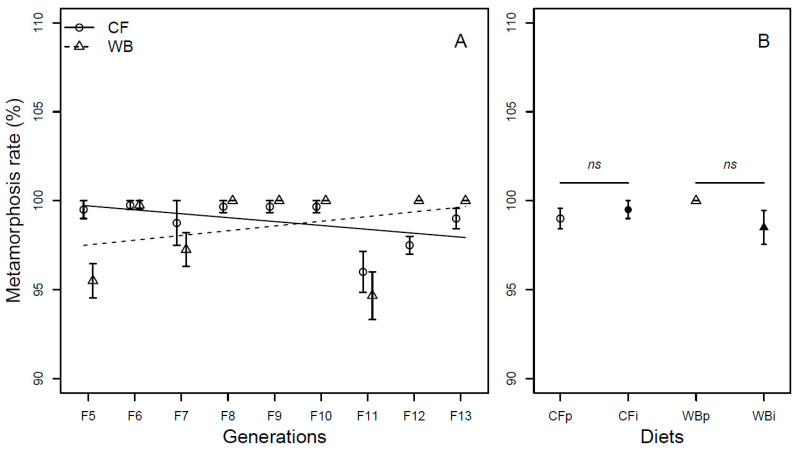
Metamorphosis rate (mean ± sem) of BSF (%) reared on high-quality CF diet and single-sourced, low-quality WB diet. Panel A: Responses to multiple subsequent generations. The solid black (CF) and dashed (WB) lines indicate the fitted model of the data (**A**). Panel B: Responses to the interchanged dietary experiment in F13 (**B**), where the parent dietary treatments CFp and WBp were compared to the interchanged dietary treatments CFi and WBi. Thus, “WBi” refers to animals raised on the WB diet after being raised for 12 generations on the CF diet. Lines, indicate pairwise comparisons showing no significant (ns) differences.

**Figure 7 insects-14-00821-f007:**
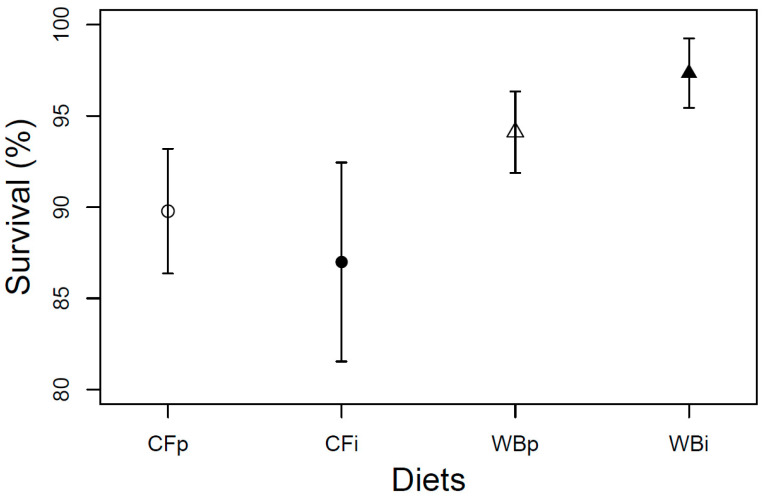
Larval survival (mean ± sem) of BSFL at harvest (%) reared on the parent and interchanged CF and WB diets during the interchanged dietary experiment in F13. “WBi” refers to animals raised on the WB diet after being raised for 12 generations on the CF diet.

**Figure 8 insects-14-00821-f008:**
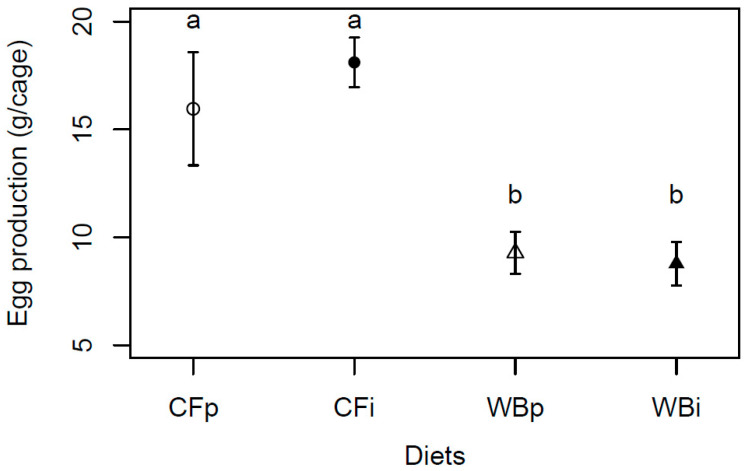
Egg production (mean weight ± sem) of BSFL (g/cage) reared on the parent and interchanged CF and WB diets during the interchanged dietary experiment in F13. “WBi” refers to animals raised on the WB diet after being raised for 12 generations on the CF diet. Contrasts with different letters are significantly different (Tukey test, *p* < 0.05).

**Table 1 insects-14-00821-t001:** Composition of WB and CF substrates in terms of dry matter (DM), ash, macronutrient (crude protein, lipid, and carbohydrate) and essential amino acid contents. * The tryptophan content was not analyzed in the WB substrate.

	CF	WB
DM (%)	30.50	29.13
Ash content (%)	6.02	5.60
Crude protein (% DM)	21.02	17.00
Crude lipid (% DM)	10.65	10.31
Carbohydrate (%DM)	62.31	67.09
Lysine (% DM)	1.28	0.46
Threonine (% DM)	0.81	0.45
Isoleucine (% DM)	0.87	0.42
Leucine (% DM)	1.63	0.86
Histidine (% DM)	0.54	0.34
Phenylalanine (% DM)	1.06	0.58
Valine (% DM)	1.02	0.60
Arginine (% DM)	1.34	0.75
Methionine (% DM)	0.43	0.17
Tryptophan (% DM)	0.27	NA *

## Data Availability

The datasets from the current study are available from the corresponding author upon reasonable request.

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
