# Peer review of "Rapid Evolutionary Adaptation to Diet Composition in the Black Soldier Fly (Hermetia illucens)"

_insects, 2023, doi:10.3390/insects14100821_

Round 1

Reviewer 1 Report

Good paper, interesting 

Author Response

Thank you!

Reviewer 2 Report

This paper seeks to exploit genetic variation of BSF to produce specialist lines adapted to low quality substrates, which normally produce suboptimal results in BSF rearing. The importance of such work to the BSF industry is immense.

I appreciate Figure 1, the flowchart, very much.
The sample size strikes me as more than sufficient for this work.

I am a native English speaker, and the language level of this paper is excellent. I did not spot many typos or severe grammatical problems.

For Figures 2 and onward, it is not immediately clear what parent an interchanged treatments are. I would recommend editing the caption to Figure 2 at least by adding the text "as described in Figure 1" or "(see Figure 1)" right after you describe CFi and WBi.

For these figures where you compare CFp to CFi and WBp to WBi, is it possible to add a bar and symbols comparing the CF's to the WBs? For example, it looks highly significant in Figure 2B and likely insignificant in Figure 6B, but it would be good to have the pairwise comparisons, perhaps like in Figure 8.

It is striking that larval survival was highest when reared on wheat bran (the low quality diet) regardless of what food they had been reared on generations prior.

The one sentence I am unsure about is: "Thus, the adaptive response in the WB lines was not in the form of specialization, but rather in the form of a more general high performance." Is that the case? If the WB lines were becoming more high performing, then the performance of CFi and WBp should be similar and also higher than CFp and certainly WBi. It is hard to confirm that from your data because you did not do pairwise comparisons between CFi and WBp except in egg production, at least not according to the figures. I would like to see this idea defended better, if the data does support it.

I would also like some discussion of where the field should go from here. How many generations before specialization or at least adaptation into a strain well-suited for a particular substrate would occur? Does the data suggest single, low-quality feeds are not necessarily that bad? What experiments would you like to see done from here?

Author Response

This paper seeks to exploit genetic variation of BSF to produce specialist lines adapted to low quality substrates, which normally produce suboptimal results in BSF rearing. The importance of such work to the BSF industry is immense.

I appreciate Figure 1, the flowchart, very much.
The sample size strikes me as more than sufficient for this work.

Thank you!

I am a native English speaker, and the language level of this paper is excellent. I did not spot many typos or severe grammatical problems.

Thank you!

For Figures 2 and onward, it is not immediately clear what parent an interchanged treatment are. I would recommend editing the caption to Figure 2 at least by adding the text "as described in Figure 1" or "(see Figure 1)" right after you describe CFi and WBi.

The recommendation was addressed, and an additional text was added to the figure legend to describe the treatments.  

For these figures where you compare CFp to CFi and WBp to WBi, is it possible to add a bar and symbols comparing the CF's to the WBs? For example, it looks highly significant in Figure 2B and likely insignificant in Figure 6B, but it would be good to have the pairwise comparisons, perhaps like in Figure 8.

The comparison between CF and WB treatments was already documented with a stronger data set across generations (panel A). As the obvious treatment differences were maintained, we prefer to restrict analysis to the a priori pairwise comparisons. In this way we avoid losing statistical power.

It is striking that larval survival was highest when reared on wheat bran (the low-quality diet) regardless of what food they had been reared on generations prior.

Although there was no significant difference in survival rate of BSFL feed  CF or WB diets, we have added a potential explanation for this interesting tendency. We believe this is related to structural differences of the two diets.

The one sentence I am unsure about is: "Thus, the adaptive response in the WB lines was not in the form of specialization, but rather in the form of a more general high performance." Is that the case? If the WB lines were becoming more high performing, then the performance of CFi and WBp should be similar and also higher than CFp and certainly WBi. It is hard to confirm that from your data because you did not do pairwise comparisons between CFi and WBp except in egg production, at least not according to the figures. I would like to see this idea defended better, if the data does support it.

We agree and have revised the section to avoid the undocumented speculation on specialization vs general high performance.

I would also like some discussion of where the field should go from here. How many generations before specialization or at least adaptation into a strain well-suited for a particular substrate would occur? Does the data suggest single, low-quality feeds are not necessarily that bad? What experiments would you like to see done from here?

The field is still in its infancy, and we can’t sensibly provide concrete answers on the numbers of generations needed for adaptation, although we expect this to be a slow process. Thus, we have included the following sentences in the discussion: “Future studies integrating classical breeding, genetic information and phenotypic evaluation can provide an avenue to explore the potential for improving performance on low-quality diets in insect production. Similarly, new studies where candidates are not selected based on phenotypes but on breeding values (the genetic merit of an individual) will likely be more efficient leading to faster selection responses and this also allows for control of inbreeding rates and simultaneous selection on multiple traits. Traits related to performance on low-quality food are likely highly multi-factorial and covered by many genes with small effect. Thus, selection responses are expected to be slow and other studies on important model and agricultural species have revealed continued responses to selection and no signs of reaching plateaus even after >100 generations of selection.”

Reviewer 3 Report

Manuscript insects-2575301 by Gligorescu et al. reports results of a laboratory selection for a better adaptation to a low-quality food substrate by black soldier flies. The study was properly designed and executed. The manuscript is written clearly and is easy to understand. However, there are several issues that, in my opinion, need to be addressed before publication.

Lines 45-46. What is the definition of a complex diet?

Lines 57-60. Does simplicity always imply low quality?

Line 89. What does metamorphosis mean in this context?

Line 126. What does a replicate line mean? There were two treatments (diets) and four replications.

Lines 140-141. What was the purpose of sorting juveniles by size? What happened to the juveniles of a different size that could get through the mesh?

Lines 166-167. This is unclear.

Lines 198-201. This is unclear. Was metamorphosis calculated as a percentage of empty puparia out of the total puparia?

Line 215. Why were generations treated as a continuous variable? Generations were discrete, each starting with an egg and ending with an adult.

Lines 231-237. This is redundant with the previous section.

Lines 239-241 and 311-313. Since the interaction between diet and generation was not significant, it is not appropriate to discuss generational trends separately for the two diets.

Line 366. Evolutionary change usually implies that the change is hereditary. Was this the case in the present study? This issue needs to be discussed.

Lines 399-400. Alternatively, the change may have been ongoing for the 40 generations in culture.

Legends for Figures 2-8. Treatment abbreviations need to be defined.

Author Response

Manuscript insects-2575301 by Gligorescu et al. reports results of a laboratory selection for a better adaptation to a low-quality food substrate by black soldier flies. The study was properly designed and executed. The manuscript is written clearly and is easy to understand. However, there are several issues that, in my opinion, need to be addressed before publication.

Thanks for the thorough comments

Lines 45-46. What is the definition of a complex diet?

We have now better-defined complex diets as being composed of several ingredients to ensure high and balanced nutritional value.

Lines 57-60. Does simplicity always imply low quality?

We do not imply that simplicity always leads to low-quality and do not intend to say so. However, in line with the argument for complex diets, simple diets are more likely to be nutritional imbalanced and lack macro and micronutrients and therefore being of inferior quality.

Line 89. What does metamorphosis mean in this context?

The metamorphosis was changed with metamorphosis rate in line 89 as well as throughout the text.

Line 126. What does a replicate line mean? There were two treatments (diets) and four replications.

We have rephrased the sentence to specify the meaning of replicate lines.

Lines 140-141. What was the purpose of sorting juveniles by size? What happened to the juveniles of a different size that could get through the mesh?

We have adjusted the text and better explain the rational behind sorting for similar size, which was to synchronize the development stage.

Lines 166-167. This is unclear.

We have reformulated the sentence to better explain the procedure.

Lines 198-201. This is unclear. Was metamorphosis calculated as a percentage of empty puparia out of the total puparia?

Yes, and we have now corrected this in the manuscript, thank you.

Line 215. Why were generations treated as a continuous variable? Generations were discrete, each starting with an egg and ending with an adult.

In this analysis, generations (which are discrete in principle) represent time, which is continuous. Using a continuous approach allow for much more powerful and general and biological meaningful interpretation of the result (as all data are analyzed together and general patterns rather than random variation among single sets of data points are in focus).

Lines 231-237. This is redundant with the previous section.

Since this is redundant, we decided to remove this paragraph.

Lines 239-241 and 311-313. Since the interaction between diet and generation was not significant, it is not appropriate to discuss generational trends separately for the two diets.

We agree that there are no significant differences between the responses of the two diets in these two cases. However, we do report model estimates (slopes) from the model – just as we report the P-values from the evaluation of the model. Both types of estimates can be informative and should be reported. We have carefully checked that we do not claim differences that are not supported by the tests.

Line 366. Evolutionary change usually implies that the change is hereditary. Was this the case in the present study? This issue needs to be discussed.

This is a key question, and the main point of the interchanged diet experiment. We performed this experiment to test whether trait differences were maintained (evolutionary change) or dependent on the immediate environment (i.e. diet experienced in the given generation). While we as expected found that diet quality is the main driver of performance on the measured traits, we also saw some trait changes maintained in the lines reared for multiple generations on low quality diet, which we interpret as inherited adaptation. Upon inspection of the text, we have not found a real need to inflate the discussion on this matter.

Lines 399-400. Alternatively, the change may have been ongoing for the 40 generations in culture.

This would not really explain the differences among the two diet treatments, as all lines for both regimes were established simultaneously from the main culture.

Legends for Figures 2-8. Treatment abbreviations need to be defined.

As also suggested by another referee we have specified the treatment better in all legends.

Reviewer 4 Report

This is an interesting study with results that will be of interest to other researchers in this field and to commercial businesses.

There are a few minor typos to be addressed and the authors should also consider the following:

Can any comment be made on the sex ratio in the mating cages for each generation. Could this have changed and could this contribute to the difference in egg production between diets at F13?

Figure 1 Would it be possible to make the image larger so that the text is easier to read and reduce the wording in the description?

For all figures - could the A and B be placed outside of the panels to make this more visible.

Author Response

This is an interesting study with results that will be of interest to other researchers in this field and to commercial businesses.

Thank you for the interest in our work.

There are a few minor typos to be addressed and the authors should also consider the following:

We have proof read the text to the best of our ability

Can any comment be made on the sex ratio in the mating cages for each generation. Could this have changed, and could this contribute to the difference in egg production between diets at F13?

Yes, we have now included a comment on the sex ratio and explain why this was not further addressed. Overall, very little deviation from 1:1 sex ratio was seen.

Figure 1 Would it be possible to make the image larger so that the text is easier to read and reduce the wording in the description?

Yes, we have now enlarged the figure.

For all figures - could the A and B be placed outside of the panels to make this more visible.

We have not placed A and B outside the figure, but we have enlarged the figures to be more visible.